# High-throughput DNA extraction and cost-effective miniaturized metagenome and amplicon library preparation of soil samples for DNA sequencing

Thomas Bygh Nymann Jensen●[◉], Sebastian Mølvang Dall[◉], Simon Knutsson, Søren Michael Karst, Mads Albertsen●*

Center for Microbial Communities, Dept. of Chemistry and Bioscience, Aalborg University, Aalborg, Denmark

◉ These authors contributed equally to this work.
* ma@bio.aau.dk, madsalbertsen85@gmail.com

**Data Availability Statement:** All sequencing data was uploaded to the ENA database under accession number PRJEB65366. Scripts for

## Abstract

Reductions in sequencing costs have enabled widespread use of shotgun metagenomics and amplicon sequencing, which have drastically improved our understanding of the microbial world. However, large sequencing projects are now hampered by the cost of library preparation and low sample throughput, comparatively to the actual sequencing costs. Here, we benchmarked three high-throughput DNA extraction methods: ZymoBIOMICS™ 96 MagBead DNA Kit, MP Biomedicals™ FastDNA™-96 Soil Microbe DNA Kit, and DNeasy® 96 PowerSoil® Pro QIAcube® HT Kit. The DNA extractions were evaluated based on length, quality, quantity, and the observed microbial community across five diverse soil types. DNA extraction of all soil types was successful for all kits, however DNeasy® 96 PowerSoil® Pro QIAcube® HT Kit excelled across all performance parameters. We further used the nanoliter dispensing system I.DOT One to miniaturize Illumina amplicon and metagenomic library preparation volumes by a factor of 5 and 10, respectively, with no significant impact on the observed microbial communities. With these protocols, DNA extraction, metagenomic, or amplicon library preparation for one 96-well plate are approx. 3, 5, and 6 hours, respectively. Furthermore, the miniaturization of amplicon and metagenome library preparation reduces the chemical and plastic costs from 5.0 to 3.6 and 59 to 7.3 USD pr. sample. This enhanced efficiency and cost-effectiveness will enable researchers to undertake studies with greater sample sizes and diversity, thereby providing a richer, more detailed view of microbial communities and their dynamics.

## Introduction

The drastic reduction in sequencing costs has enabled more researchers to utilize next-generation sequencing in their field of study [1]. In the field of microbial ecology especially, the reduced costs have enabled an increase in the scope of the projects, as thousands of samples

processing data and generating figures can be found on GitHub https://github.com/SebastianDall/MFD_HT_PAPER.

**Funding:** The study was conducted as part of the MicroFlora Danica project awarded to MA by the Poul Due Jensen Foundation (https://www.pdjf.dk/en/). The funders had no role in study design, data collection and analysis, decision to publish, or preparation of the manuscript.

**Competing interests:** The authors have declared that no competing interests exist.

are required to fully understand the diversity of microbial ecosystems [2–5]. The ongoing reductions in sequencing costs means that large sequencing projects are now cost-limited by the cost associated with hands-on time and sample preparation. However, new automated or semi-automated workflows utilizing liquid handlers and drop dispensing technology seem promising regarding the reduction of both labor time and reaction volumes–ultimately reducing the overall costs [1, 6–8].

Soil samples are especially problematic in high-throughput (HT) DNA extraction workflows due to the range of soil physical and chemical properties [1, 9, 10]. Furthermore, the majority of the proposed protocols are not easy to convert to a HT format due to steps that are either difficult to automate, time-consuming, or include hazardous substances. Although several commercial soil-specific HT DNA extraction kits are available (Table 1), these have not been independently tested on a diverse range of soil types.

With the reduction in sequencing costs, library preparation has become a significant proportion of total project costs. One of the first kits for cost-effective next-generation sequencing of low input material, which allowed for multiplexing of many samples, was the Nextera XT DNA library preparation kit [43–45]. The Nextera XT library preparation protocols for small genomes, PCR amplicons, plasmids, or cDNA have undergone several transformations since the first release in 2012. The first Nextera XT protocols were easy to use, however the expensive reagents were limiting the scale of sequencing projects [46]. To reduce the library preparation costs, earlier work focused on diluting the expensive reagents or replacing them with cheaper alternatives [47], however in 2017 the Nextera Flex (later renamed to Illumina DNA prep) protocol was introduced. The Illumina DNA prep utilizes bead-linked transposases, rendering the previous cost-effective protocols less useful. Previous work has shown it is also possible to dilute the reagents in the Nextera Flex kit [46], however another strategy for reducing the overall costs without tampering with the reagents is through miniaturization.

Here we present and benchmark a complete HT workflow from DNA extraction to miniaturized Illumina amplicon or metagenome library (Fig 1).

## Results

### DNA extraction benchmark

The three HT DNA extraction kits were benchmarked on five different soil types (S1 Table). PowerSoil Pro HT and ZymoMagbead HT both come without a lysing matrix, whereas FastSpin HT, FastSpin LT, and PowerSoil LT do. To better evaluate the DNA extraction chemistry of the HT kits, it was decided to employ the lysing matrix E from FastSpin LT. DNA extraction with PowerSoil LT was performed with its native lysing matrix. Samples were bead-beaten for a total of six minutes at 1800 RPM when using lysing matrix E. All samples were prepared according to the manufacturer's protocol.

Generally, the kits were able to extract DNA from all investigated soil types; however low amounts were extracted from Beach Sand, which was likely due to low biomass relative to the other soil types (S2 Table). PowerSoil Pro HT extracted more DNA than both FastSpin HT and ZymoMagbead HT ($p < 0.001$, ANOVA on ranks, n = 45). The DNA yield of PowerSoil Pro HT and PowerSoil LT were comparable (p = 0.08, ANOVA on ranks, n = 30). The FastSpin LT DNA yields could not be determined due to unreliable Qubit measurements likely caused by a high concentration of residual humic substances after DNA extraction (S1 File).

Generally, the 260/280 ratios were $\sim 1.8$, which is considered pure for DNA. The FastSpin HT had high 260/280 ratios across all soil types, which could indicate contamination with RNA. The 260/230 ratio varied greatly between kits. Low values were measured for FastSpin LT, FastSpin HT, and ZymoMagbead HT indicating the inability to remove sample or kit

**Table 1. Commonly used and commercially available kits for DNA extraction from soil.** Extraction kits in **bold** were compared. High-throughput equipment refers to available automated solutions for the high-throughput solution.

| Manufacturer | Low-throughput Kit | High-throughput Kit | High-throughput Equipment | Reference |
|---|---|---|---|---|
| QIAGEN (previously MoBio) | **DNeasy® PowerSoil® Kit\*** **(PowerSoil LT)** | DNeasy® PowerSoil® HTP 96 Kit | | **LT:** [10–21] **HT:** [22] |
| | | MagAttract® PowerSoil® Pro DNA Kit | KingFisher® Flex/Duo epMotion® 5075 TMX | **HT:** [5, 22, 23] |
| | DNeasy® PowerSoil® Pro Kit | **DNeasy® 96 PowerSoil® Pro QIAcube® HT Kit (PowerSoil Pro HT)** | QIAcube® HT System | **HT:** [24–26] |
| MP Biomedicals | **FastDNA™ SPIN kit for Soil (FastSpin LT)** | **FastDNA™-96 Soil Microbe DNA extraction Kit (FastSpin HT)** | | **LT:** [10, 19, 20, 27, 28] |
| Macherey-Nagel | NucleoSpin™ Soil Kit | NucleoSpin™ 96 Soil Kit | | **LT:** [21, 29–32] **HT:** [33–36] |
| Zymo Research | ZymoBIOMICS® DNA Micro/Mini/Midi Kit | ZymoBIOMICS® 96 DNA Kit | | **LT:** [16, 17] **HT:** [37] |
| | | **ZymoBIOMICS® 96 MagBead DNA Kit (ZymoMagbead HT)** | Hamilton Microlab® STAR | **HT:** [22, 23] |
| | ZymoBIOMICS® Quick-DNA Fecal/Soil Microbe Micro/Mini/Midi Kit | Quick-DNA Fecal/Soil Microbe 96 Kit | | |
| | | Quick-DNA Fecal/Soil Microbe 96 Magbead Kit | Hamilton Microlab® STAR | **HT:** [38] |
| Omega Bio-Tek | E.Z.N.A.® Soil DNA Kit | Mag-Bind® Environmental DNA 96 Kit | Hamilton Microlab® STAR Hamilton Microlab® NIMBUS KingFisher™ BioSprint® MagMAX® 96 | **LT:** [21, 39–42] |

*QIAGEN has replaced this kit with DNeasy PowerSoil Pro. C2 and C3 have been replaced by CD2 which is a combination of the two buffers.

LT: Low-throughput, HT: High-throughput.

contaminants absorbing at 230 nm (S2 Table). The PowerSoil Pro HT and PowerSoil LT had 260/230 ratios closest to that of pure nucleic acids (2.0–2.2). The 260/230 ratio of PowerSoil Pro HT was not significantly different from its LT counterpart (Mann-Whitney U, p = 0.35), but a higher 260/280 ratio was observed (Mann-Whitney U, p<0.001).

DNA shearing among the HT kits was highest for PowerSoil Pro HT with a mean peak DNA fragment length of 7.3 kb (n = 15, sd = 0.7 kb) compared to an average length of 13.2 kb and 16.0 kb for FastSpin HT (n = 15, sd = 3.9 kb) and ZymoMagbead HT (n = 15, sd = 5.4 kb), respectively (S1 Fig). DNA shearing was primarily driven by the extraction kit (81.7% variance, p<0.001, ANOVA on ranks n = 75) but also the interaction of the extraction kit and soil type (10.8% variance, p<0.001, ANOVA on ranks n = 75). The mean peak fragment length of the FastSpin LT kit, 5.6 kb (n = 15, sd = 0.9 kb), was even lower than for PowerSoil Pro HT. PowerSoil LT mean peak fragment length, 8.6 (n = 15, sd = 1.4 kb), was significantly larger than that of the PowerSoil Pro HT kit (Mann-Whitney U, p<0.001).

Miniaturized Illumina amplicon libraries were sequenced and processed with a standard bioinformatic pipelines (see methods). Two out of three Organic soil amplicon libraries failed for FastSpin LT and ZymoMagbead HT. Both kits resulted in low 260/230 ratios suggesting the presence of contaminants. Amplicon libraries were successfully sequenced for all soil types for both FastSpin HT and Powersoil Pro HT; however, with both methods one library from

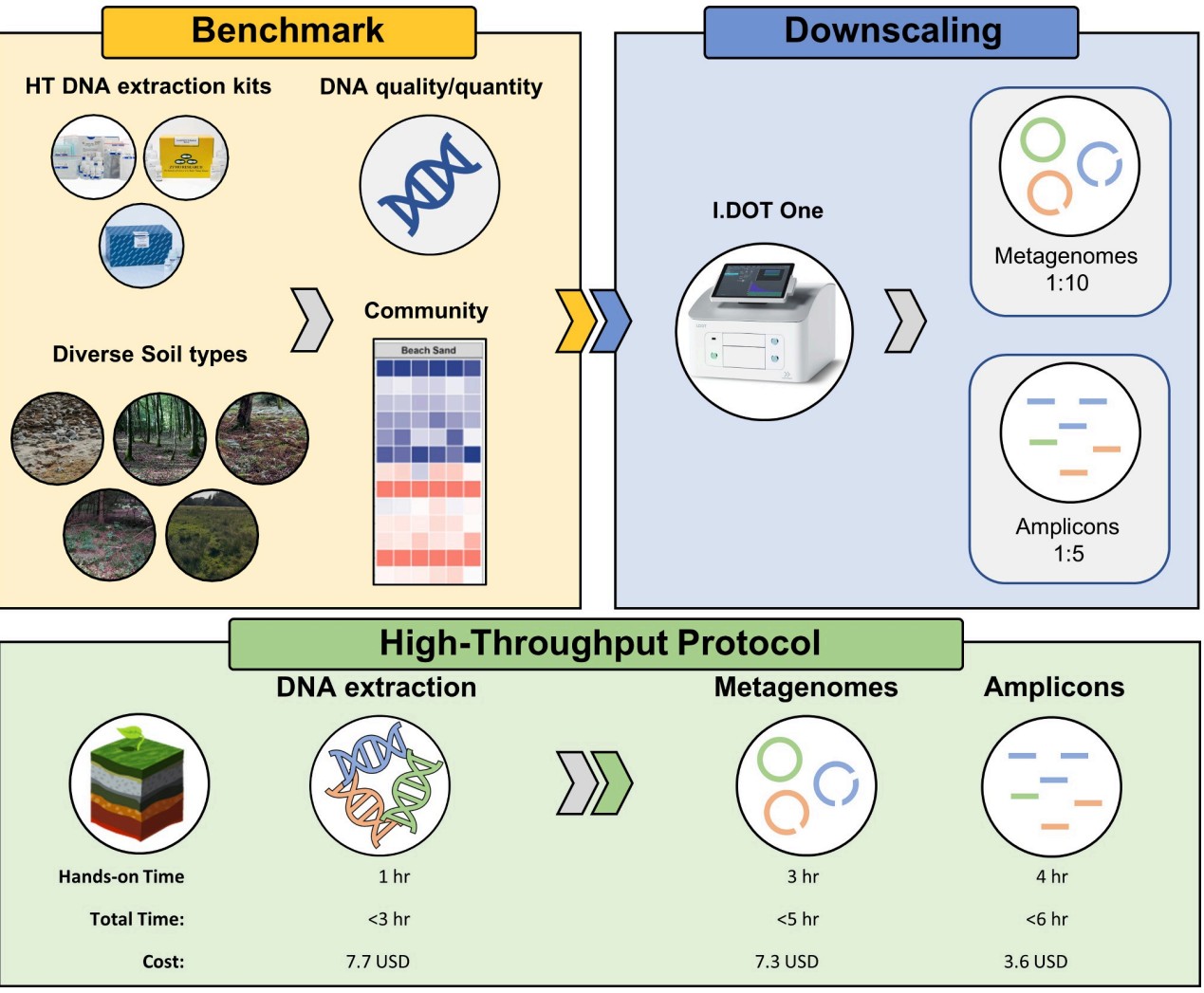

**Fig 1. Experimental design.** Top: Experimental design. Five different soil types were used to benchmark three different HT DNA extraction methods. DNA extractions were evaluated by DNA quality, quantity, length, and observed community profile. The I.DOT One was subsequently used to miniaturize metagenomes and amplicons. Bottom: Hands-on time, total time, and cost associated with each step from DNA extraction to prepared metagenome or amplicon libraries. The time reflects the processing time of a full 96-well plate, whereas costs are calculated per sample. For both sequencing strategies, two 96-well plates can be processed concurrently with only a minor increase in total time.

the Sand samples was removed as an outlier evaluated by principal component analysis (PCA)–most likely due to cross-contamination of the samples.

Based on ANOVA, the Shannon diversity index was significantly different for both soil type (83.2% variance explained, p<0.001), DNA extraction kit (7.1% variance explained, p<0.001), and the interaction of soil type and DNA extraction kit (6.7% variance explained, p<0.001). The difference between PowerSoil Pro HT and FastSpin HT was 0.04 (Tukey's HSD, p = 0.03), between FastSpin HT and ZymoMagbead was 0.05 (Tukey's HSD, p = 0.02), and between PowerSoil Pro HT and ZymoMagbead HT was 0.09 (Tukey's HSD, p<0.001). Similarly, Bray-Curtis dissimilarity was significantly different for both soil type (61.5% variance explained, p<0.001), for the extraction kit (6.1% variance explained, p<0.001), and the interaction of soil type and DNA extraction kit (23.2% variance explained, p<0.001).

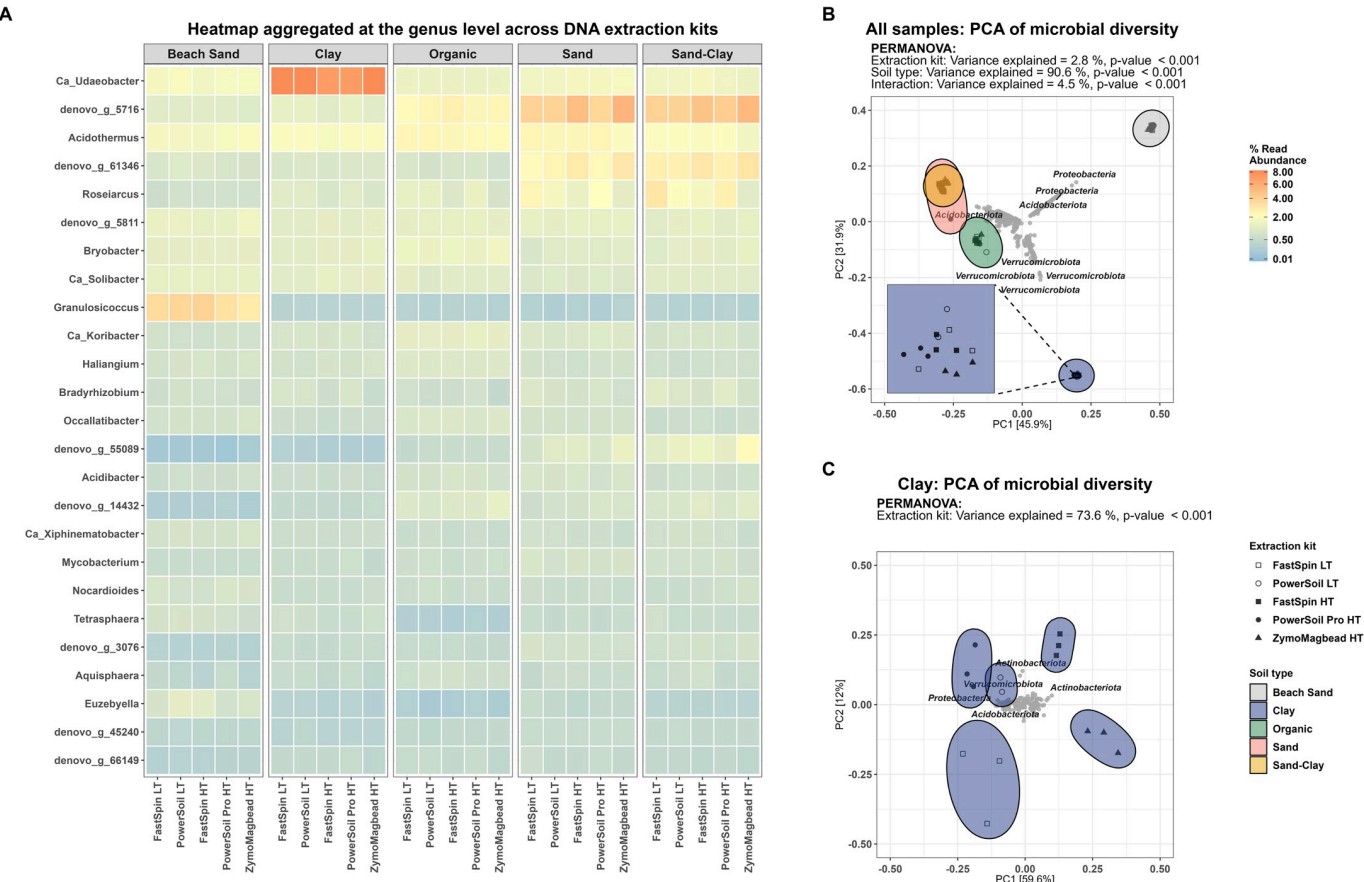

**Fig 2. Community characteristics for DNA extraction kits.** (A) Heatmap of community profile at phylum level across DNA extraction kits faceted by soil type. (B) PCA on Hellinger transformed relative abundance. (C) PCA on Hellinger transformed relative abundance for the Clay samples only. For all plots: ASVs not exceeding 0.1% relative abundance in at least one sample were filtered out.

The microbial community profiles were similar across all kits (Fig 2A). PCA revealed the communities clustered according to soil type, not DNA extraction kit. Based on a PERMA-NOVA 2.8% variance was explained by DNA extraction kit (p<0.001) and 90.6% by soil type (p<0.001) (Fig 2B). When stratifying for soil type PCA revealed samples clustered by DNA extraction kit (Figs 2C and S2).

Based on the DNA extraction characteristics, the high amplicon library success rate for all soil types, and the consistent community profile the PowerSoil Pro HT DNA extraction kit was selected for further optimization. Specifically, the effects of bead-beating time and intensity on the observed community structure as well as DNA quantity and length were investigated. Both bead-beating time and intensity affected the DNA yield and observed microbial community, however, little difference was observed between six minutes of bead-beating at 1600 RPM and 1800 RPM. Increasing the bead-beating intensity to 1800 RPM did increase shearing, why bead-beating for a total of six minutes at 1600 RPM was chosen (S2 File). Reducing the input amount from 125 mg to 50 mg had no effect on the observed microbial community (S2 File). The PowerSoil Pro HT kit can furthermore be semi-automated with the QIAcube HT system to reduce the hands-on time. In total, the DNA extraction cost per sample based on chemicals and disposables for PowerSoil Pro HT was 7.7 USD.

## Miniaturized Illumina amplicon library protocol

Amplicon libraries were successfully prepared for all soil types (S3 Table). Shannon diversity index was not significantly affected by the library volume (0.3% variance explained, p = 0.33, ANOVA, n = 30) when blocking the contribution from soil type (p<0.001), and the interaction between soil type and library volume (p = 0.04). The data violate the assumption of normal distribution but not the assumption of heteroscedasticity for both grouping factors. ANOVA based on ranks did yield very similar results, but it changed the p-value of the interaction to p = 0.14.

Mean Bray-Curtis dissimilarity between replicates was not significantly affected by library volume (p = 0.86, ANOVA on ranks, n = 30). Furthermore, the mean Bray-Curtis dissimilarity between protocols was similar to that observed among the replicates, except in the case of Beach Sand (p = 0.002, Mann-Whitney U = 15, Bonferroni correction for multiple testing).

Community profiles at the genus level were similar between standard and miniaturized amplicon libraries (Fig 3A). When conducting a differential abundance analysis of all ASVs between the standard and miniaturized protocol a total of 19 ASVs were found to be differentially abundant (Fig 3B). In total, 11 of these (8 from Beach Sand and 3 from Organic) were completely absent across all replicates in the miniaturized protocol, likely showcasing the limitations of miniaturization for low biomass and/or highly diverse samples. No differential abundant ASVs were detected in Sand, Sand-Clay, and Clay (S3 Fig).

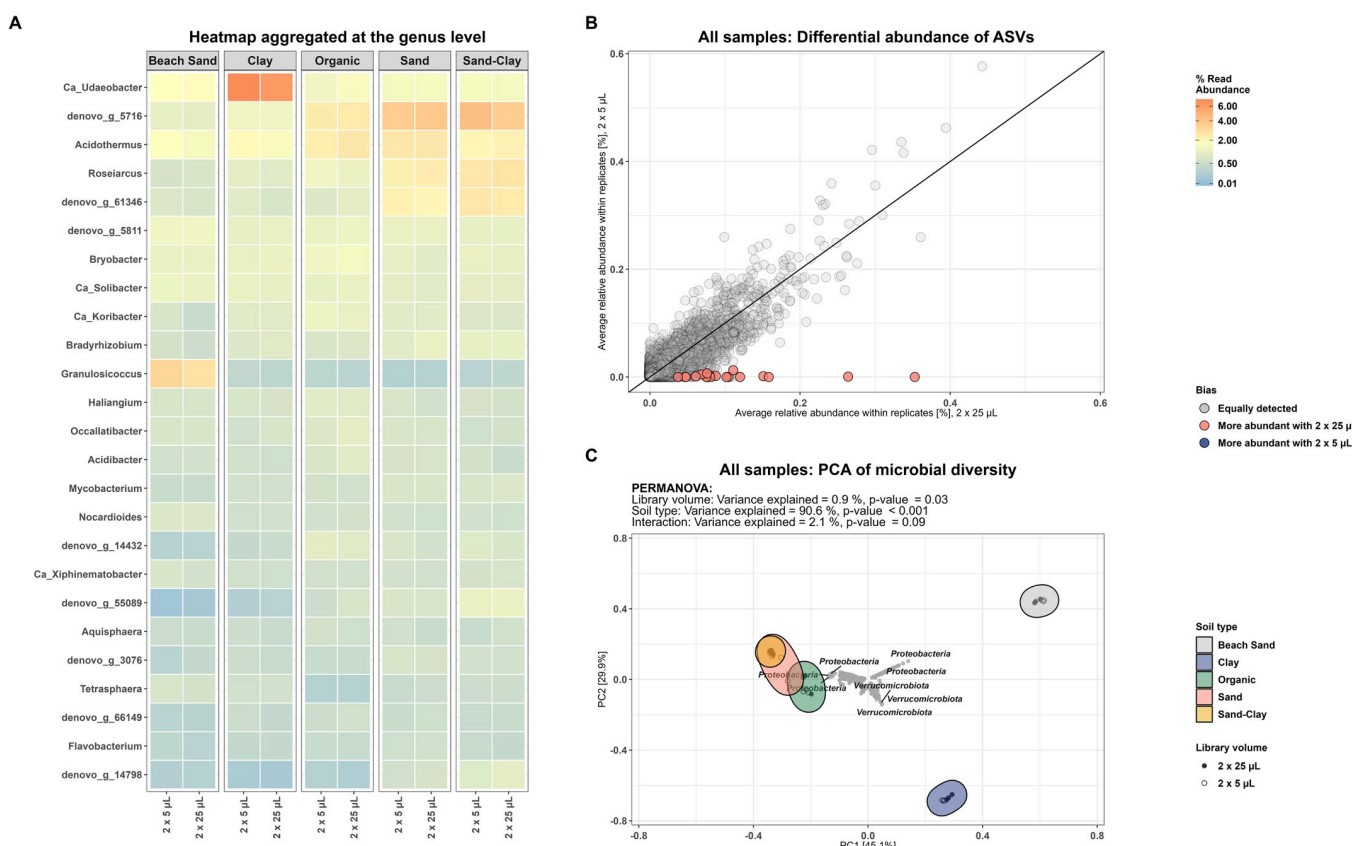

**Fig 3. Comparison of community characteristics between standard and miniaturized amplicons.** (A) Heatmap of community profile at the genus level across reaction volume faceted by soil type. (B) Differential relative abundance plot of ASVs. (C) Hellinger transformed relative abundance, PCA. ASVs not exceeding 0.1% relative abundance in at least one sample were filtered out.

Using only ASVs with a relative abundance above 0.1% the variance explained by the different library preparation protocols amounts to 0.9% of the total variance observed (PERMANOVA, p = 0.03) (Fig 3C). PERMANOVA showed no significance for library volume when performed per soil type (S4 Fig).

The miniaturized protocol effectively reduced our per sample chemical and plastic cost for amplicon library preparation from 4.9 USD to 3.6 USD.

## Miniaturized Illumina DNA prep protocol

Illumina DNA prep libraries were successfully prepared and sequenced for all soil types with both the standard and the miniaturized protocol (S4 Table) except one library for the Clay samples using the miniaturized protocol.

All reads identified as 16S fragments were aggregated to the genus level (see methods). Based on ANOVA on ranks, Shannon diversity index was not significantly affected by the library volume (0% variance explained, p = 1). Soil type explained 80.2% variance (p<0.001). From the ANOVA analysis based on ranks, the mean Bray-Curtis dissimilarity between replicates was affected by the library volume (6.8% variance explained, p = 0.02) and was slightly lower in the miniaturized protocol (TukeyHSD, p = 0.02). Soil type again accounted for the largest proportion of the variance (65% variance explained, p<0.001). The mean Bray-Curtis dissimilarity for replicates within protocols was not significantly different to the observed dissimilarity of replicates between protocols for any soil type (S4 Table).

Comparison of the relative abundance at the genus level revealed very similar profiles between the standard and miniaturized protocol, regardless of soil type (Fig 4A). None of the identified genera were found to be differential abundant between the protocols (Figs 4B and S5). Using only genera with a relative abundance above 0.1% the variance explained by the different library preparation protocols amounts to 0.3% of the total variance observed (PERMANOVA, p = 0.21) (Fig 4C). When performing PERMANOVA per soil type no significance for library volume was observed (S6 Fig).

As part of the optimization step of the protocol, different ratios of Sample Purification Beads (SPB/IPB) were tested to obtain an optimal fragment size distribution, as a shift towards shorter fragments was observed with the miniaturized protocol (S3 File). Miniaturizing the Illumina DNA library prep with a factor of 10 reduced our per sample chemical and plastic cost for preparation of metagenomic libraries from 52.2 USD to 7.3 USD.

## Discussion

Microbial communities are often entangled with questions which require large sample sizes to answer. Hence, to facilitate this it is paramount that sample preparation is converted to a HT setting, and sample preparation costs are significantly reduced to enable more large scope projects [2]. For soil samples, the range of physical and chemical properties pose a problem for DNA extraction kits, as the kit often needs to be optimized for each sample type, which is infeasible in a HT setting. In this benchmark, PowerSoil Pro HT slightly outperformed other HT kits for all investigated soil types based on DNA quality, specifically the 260/230 ratio. Another advantage of the PowerSoil Pro HT was the semi-automated QIAcube extraction protocol, which significantly reduces hands-on time and the bias associated with manual liquid handling. Amplicon libraries for all soil types were successfully sequenced with DNA from the PowerSoil Pro HT (n = 15) protocol and FastSpin HT (n = 15). PowerSoil Pro HT did on average fragment the DNA more than other protocols, which can be a disadvantage for long-read DNA sequencing technologies, such as Oxford Nanopore Technology and PacBio. Though the

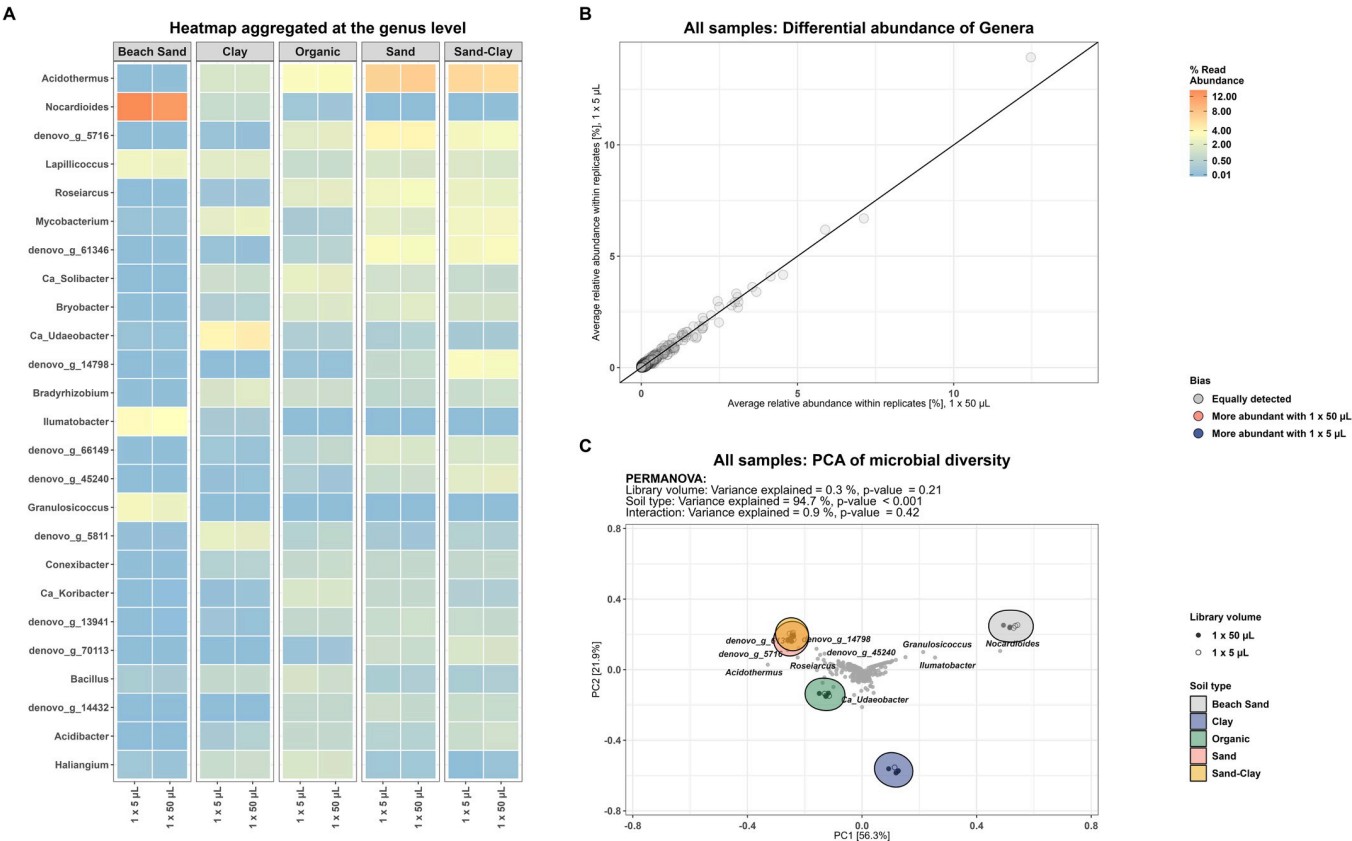

**Fig 4.** (A) Heatmap of community profile at the genus level across reaction volume faceted by soil type. (B) Differential abundance plot of all genera (C) Hellinger transformed relative abundance, PCA. Genera which did not exceed 0.1% relative abundance in at least one sample were filtered out.

peak fragment size was longer than the golden threshold of 7 kb [48] for all soil types most of the DNA was below this threshold.

The microbial community could successfully be analyzed with miniaturized reaction volumes for both amplicons and metagenomes. Metagenomes could be prepared in a 1:10 reaction volume, whereas amplicon reaction volumes could be miniaturized with a factor of five. A downside to the miniaturized reaction volumes is the entry cost of the nano-liter drop-dispensing platforms, ranging from 100.000 to 300.000 USD, as well as expensive servicing fees and highly priced plastic consumables. However, in the context of large sequencing projects, the entry cost is small compared to the reduction in library preparation cost and hands-on time. In our case, the cost savings of miniaturizing the metagenomes library protocol exceeded the price of the I.DOT One after ∼2000 samples. The downscaled protocols presented here can in be performed by hand, without needing the I.DOT One. Manual pipetting of small volumes consistently, however, can be challenging.

Several steps were automated with the QIAcube® HT and I.DOT One to reduce the hands-on time. The protocol could be further improved by automating some of the liquid handling steps, especially the clean-up steps in the HT miniaturized Illumina DNA prep protocol.

The cost of sequencing will depend strongly on the research question to be answered as some projects will require more sequencing depth than others. On the Illumina NovaSeq 6000, a sequencing depth of 5 Gbp corresponds to approximately 19 USD per sample in sequencing chemical costs. For amplicons, multiplexing the maximum number of samples (384 with

current Illumina barcodes) for sequencing on the Illumina NovaSeq 6000 platform would give excessive depth even for the lowest throughput flow cell available. A single SP sequencing run would amount to ∼9–12 USD per sample in sequencing costs and would provide a theoretical median depth of 1.69 to 2.08 million ∼250–450 bp amplicons. Amplicon sequencing costs could be tremendously reduced by designing additional barcodes to allow more extensive multiplexing [46, 49]. Acquiring 10.000 unique (100 x 100) indexes of 500 picomole each, and sequencing on a SP flow cell would yield 65–80 thousand amplicons per sample, amounting to 48 cents for 250 bp amplicons and 62 cents for 450 bp amplicons.

## Conclusion

The DNeasy® 96 PowerSoil® Pro QIAcube® HT Kit chemistry outperformed both the FastDNA™-96 Soil Microbe DNA extraction Kit and the ZymoBIOMICS® 96 MagBead DNA Kit based on DNA purity and yield, though at the cost of shorter DNA. Metagenomes and amplicons could successfully be miniaturized without affecting the observed microbial community for five different soil types, effectively reducing the per sample chemical and plastic costs for library preparation to 7.3 USD and 3.6 USD, respectively.

## Materials and methods

### Soil types

Five different soil types were included. All samples consist of five topsoil subsamples (0–20 cm) taken within five areas of ∼80 m2 (5 m radius), which were mixed before pouring into a 100 mL sample container. pH was measured for each soil type by mixing 10 g of soil with 30 mL of deionized water [50]. After settling, pH was measured with [SI Analytics Lab 855]. Sample characteristics as well as geographic position can be found in S1 Table.

### DNA extraction

HT DNA extractions were performed in 1.2 mL 2D barcoded matrix tubes pre-filled with lysing matrix E (1.4 mm ceramic spheres, 0.1 mm silica spheres, and one 4 mm glass bead) from MP biomedicals (https://www.mpbio.com/bs/116984001b-lysing-matrix-e-barcoded-plate). Lysing Matrix E has previously been shown to effectively lyse both gram-positive and negative bacteria [51]. Before extraction, 100 μL (∼125 mg soil) of each soil type was transferred to a 2D barcoded Lysing Matrix E tube with a 1 mL syringe, whereafter the sample barcode was linked to the 2D extraction tube barcode with a Mirage Rack Reader (Ziath) and the software DataPaq™ (Ziath). DNA extractions with FastDNA™ SPIN kit for Soil and DNeasy® Power-Soil® Kit were performed with the available kit lysis tubes. Soil target input was 125 mg unless otherwise stated.

DNA extraction followed the manufacturer's protocol for all kits except for DNeasy® 96 PowerSoil® Pro QIAcube® HT Kit, which followed a slightly modified protocol.

### DNeasy® 96 PowerSoil® Pro QIAcube® HT Kit

DNA extraction followed a slightly modified protocol of the DNeasy® 96 PowerSoil® Pro QIAcube® HT Kit. Firstly, 500 μL CD1 was added to 125 mg of soil (unless otherwise stated), whereafter samples underwent three bead-beating cycles performed in two-minute intervals using the FastPrep96™. Between rounds of bead-beating the samples were kept on ice for two minutes. After lysis, samples were centrifuged at 3.486 x g for 10 minutes using an Eppendorf 5810 benchtop centrifuge, and 300 μL supernatant was transferred to a clean S-block containing 300 μL CD2 and 100 μL nuclease-free water (NFW) to meet the requirement of 700 μL for

the rest of the protocol. Samples were again centrifuged at 3.486 x g for 10 minutes, whereafter subsequent steps followed the manufacturer's protocol. The sample transfer step was done using the QIAcube® HT.

## V4 Amplicons: PCRBIO Ultramix

Standard amplicon libraries were prepared as one 50 μL reaction and subsequently split into two 25 μL reactions. Up to 20 ng of quality-controlled genomic DNA was used as the template. After 25 cycles of PCR (amplicon PCR) duplicate samples were pooled and cleaned using 0.8x CleanNGS sample purification beads and washed twice with 80% EtOH and eluted in NFW. Another 8 cycles of library PCR were performed on up to 10 ng of amplicon template and cleaned as previously described. Final libraries were quantified and pooled equimolarly to produce the final sequencing libraries. Quality control was performed using the Qubit 1X HS assay [Invitrogen™, Thermo Fisher] and either Genomic DNA ScreenTape or D1000 Screen-Tape [Agilent Technologies].

Miniaturized amplicon libraries were prepared with the I.DOT One as two individual 5 μL reactions by dispensing into two individual PCR plates. Each 5 μL amplicon PCR reaction consisted of 1.5 μL sample/NFW (target: 2 ng DNA), 2.5 μL PCRBIO 2x Ultra Mix, and 1 μL abV4-C tailed amplicon primer mix (2 μM, 400 nM final concentration). The subsequent 5 μL library PCR was prepared with 2 ng of cleaned PCR template in 1.5 μL/NFW (target: 2 ng DNA), 2.5 μL PCRBIO 2x Ultra Mix, and 0.5 μL adapter indexes (4 μM). After clean-up, libraries were pooled equimolarly using the I.DOT One.

A detailed protocol can be found at: https://github.com/SebastianDall/HT-downscaled-amplicon-library-protocol.

Amplicons were sequenced on the Illumina MiSeq platform. ASV abundance tables were generated by running AmpProc 5.1 (https://github.com/eyashiro/AmpProc) using the following choices: standard workflow, generate both otu and zotu tables, process only single-end reads, no primer region removal, amplicon region V4 and a version of the SILVA SSURef 99% v138.1 database [52] processed by AutoTax [53]. AmpProc is a wrapper script for running USEARCH11 [54] and downstream processing of output tables. AmpProc assigns taxonomy to ASVs by running SINTAX with the confidence cutoff set to 0.8 [55].

## Metagenomes: Illumina DNA prep

Standard metagenomic libraries were prepared according to the recommendations in the Illumina DNA prep protocol [Illumina] but eluted in NFW instead of the resuspension buffer (RSB).

Miniaturized metagenomic libraries were prepared with the I.DOT One and followed a 1:10 reagent volume reduction of the Illumina DNA prep protocol. Firstly, a 3 μL template (target: 20 ng DNA) was prepared with the I.DOT One, whereafter 2 μL BLT/TB1 master mix was added using the I.DOT One. The reaction was incubated in a thermocycler running the TAG-program from the Illumina DNA Prep protocol. The tagmentation reaction was stopped by adding 1 μL TSB using the I.DOT One, and incubation of the reaction with the PTC program in a thermocycler. After stopping the tagmentation, the libraries were washed twice with 10 μL TWB. 2 μL EPM, 2 μL NFW and 1 μL IDT® Illumina UD index was added to each well by the I.DOT One and using an epMotion® 96 [Eppendorf], respectively. Based on the original genomic DNA input, libraries were given 7 ($>$ = 4.9 ng), 8 (2.5–4.9 ng), 10 (0.9–2.5 ng), or 14 ($<$0.9 ng) cycles of the BLT-PCR program. PCR reactions were diluted with 17 μL NFW before 18 μL of the reactions were transferred to a new PCR-plate with 16 μL of sample purification beads (SPB) and 18 μL NFW in each well. After incubation, the beads were allowed to pellet before 50 μL of the supernatant was transferred to a new PCR-plate with 6 μL of SPB.

After another incubation step, the beads were washed twice with 45 μL of 80% ethanol and subsequently eluted in 20 μL NFW.

Final libraries were quantified and pooled equimolarly to produce the final sequencing libraries. Quality control was performed using the Qubit 1X HS assay [Invitrogen™, Thermo Fisher] and DS1000 or DS1000 HS ScreenTape [Agilent Technologies]. A detailed protocol can be found at https://github.com/SebastianDall/HT-Downscaled-Illumina-Metagenomes-Protocol.

Metagenome libraries were sequenced with the Illumina NovaSeq 6000 to a median depth of 5 Gbp. Raw Illumina reads were trimmed for barcodes, quality filtered, and deduplicated with fastp [56] and GNU-parallel [57], whereafter 16S rRNA genes were extracted from the quality-filtered reads. To extract the 16S reads, HMM models of the Rfam 14.7 seed alignments for Bacteria (RF00177) and Archaea (RF01959) were built [58] with hmmbuild (HMMER V3.3.2) [59]. The HMM models were used with nhmmer and seqkit [60] to extract the 16S reads which were quality filtered for the best match using the bit-score. The 16S reads were taxonomically assigned using SINTAX with the confidence cutoff set to 0.8 using the previously mentioned database. The output was transformed into an observational table and aggregated to the genus level using R. The scripts and parameters can be found at https://github.com/SebastianDall/MFD_HT_PAPER.

### Visualization and statistical analysis

All data visualization and statistical analysis were carried out in R (4.2) and Rstudio (2023.03.0 +386) with the following packages: tidyverse [61], ampvis2 [62], vegan [63], and bioconductor [64]. Source code can be found at https://github.com/SebastianDall/MFD_HT_PAPER.

### Supporting information

**S1 Fig. Genomic DNA fragment distribution across different DNA extraction kits.**
(PDF)

**S2 Fig. PCA of DNA extraction kits stratified by soil type.**
(PDF)

**S3 Fig. Differential abundance plot for each soil type.**
(PDF)

**S4 Fig. PCA of the miniaturized and standard protocol for amplicons.**
(PDF)

**S5 Fig. Differential abundance plots comparing the miniaturized and standard Illumina protocol.**
(PDF)

**S6 Fig. PCA of miniaturized and standard Illumina protocol.**
(PDF)

**S1 File. Humic substance interference of Qubit measurements.**
(PDF)

**S2 File. Optimization of PowerSoil Pro HT protocol.**
(PDF)

**S3 File. Optimization of Illumina DNA prep.**
(PDF)

**S1 Table. Sample characteristics.**
(PDF)

**S2 Table. General statistics from DNA extraction, library preparation, and community profiles based on 16S rRNA amplicon data.**
(PDF)

**S3 Table. General library characteristics of the miniaturized and standard amplicon protocol.**
(PDF)

**S4 Table. General library characteristics of the miniaturized and standard metagenome protocol.**
(PDF)

## Author Contributions

**Conceptualization:** Thomas Bygh Nymann Jensen, Sebastian Mølvang Dall, Mads Albertsen.

**Data curation:** Thomas Bygh Nymann Jensen, Sebastian Mølvang Dall.

**Formal analysis:** Thomas Bygh Nymann Jensen, Sebastian Mølvang Dall, Simon Knutsson, Søren Michael Karst, Mads Albertsen.

**Funding acquisition:** Mads Albertsen.

**Investigation:** Thomas Bygh Nymann Jensen, Sebastian Mølvang Dall, Simon Knutsson, Søren Michael Karst, Mads Albertsen.

**Methodology:** Thomas Bygh Nymann Jensen, Sebastian Mølvang Dall, Simon Knutsson, Søren Michael Karst, Mads Albertsen.

**Project administration:** Thomas Bygh Nymann Jensen, Søren Michael Karst, Mads Albertsen.

**Resources:** Mads Albertsen.

**Software:** Thomas Bygh Nymann Jensen, Søren Michael Karst.

**Supervision:** Mads Albertsen.

**Validation:** Thomas Bygh Nymann Jensen, Sebastian Mølvang Dall, Simon Knutsson, Søren Michael Karst.

**Visualization:** Thomas Bygh Nymann Jensen, Sebastian Mølvang Dall.

**Writing – original draft:** Thomas Bygh Nymann Jensen, Sebastian Mølvang Dall, Mads Albertsen.

**Writing – review & editing:** Thomas Bygh Nymann Jensen, Sebastian Mølvang Dall, Simon Knutsson, Søren Michael Karst, Mads Albertsen.

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
