## [Decision Letter · Decision Letter 0]

17 Mar 2024

High-throughput DNA extraction and cost-effective miniaturized metagenome and amplicon library preparation of soil samples for DNA sequencing

PONE-D-23-28692

Dear Dr. Albertsen,

We’re pleased to inform you that your manuscript has been judged scientifically suitable for publication and will be formally accepted for publication once it meets all outstanding technical requirements.

Kind regards,

Vittorio Sambri, M.D., Ph.D.

Academic Editor

PLOS ONE

Additional Editor Comments:

The authors do not need to address the additional reviewer's comments.

Reviewers' comments:

Reviewer's Responses to Questions

**Comments to the Author**

1. Is the manuscript technically sound, and do the data support the conclusions?

Reviewer #1: Yes

Reviewer #2: Yes

2. Has the statistical analysis been performed appropriately and rigorously? 

Reviewer #1: Yes

Reviewer #2: Yes

3. Have the authors made all data underlying the findings in their manuscript fully available?

Reviewer #1: Yes

Reviewer #2: Yes

4. Is the manuscript presented in an intelligible fashion and written in standard English?

Reviewer #1: Yes

Reviewer #2: Yes

5. Review Comments to the Author

Reviewer #1: The authors in this work compared three high-throughput DNA extraction methods on the basis of length, quality, quantity, and the observed microbial community across five diverse soil types. Amplicon and metagenomic library preparation were then miniaturized by a factor of 5 and 10, respectively, showing no significant impact on the observed microbial communities and therefore potential for a reduction in both costs and preparation times, provided one has large enough projects to offset the high entry costs.

The optimization and benchmarking of DNA extraction on a difficult starting material as diverse as soil is extremely useful.

This is a very well done work.

Reviewer #2: The authors compared three different extraction methods: the 96 MagBead 16 DNA Kit, MP BiomedicalsTM FastDNATM-96 Soil Microbe DNA Kit and DNeasy® 96 PowerSoil® Pro 17 QIAcube® HT Kit, reporting data and performance of these kits evaluated on the basis of length, quantity, quality and microbial communities observed in 5 soil types. The 260/280 and 260/230 ratios confirmed the extraction quality, which was observed to vary considerably between the kits, and confirmed the extraction quality for the DNeasy® 96 PowerSoil® Pro 17 QIAcube® HT Kit with a higher 260/280 ratio, even with a shorter average fragment length but higher extract purity.

The use of an 'I.DOT' nanolitre dispensing robot for miniaturisation was extensively demonstrated by the authors to improve quality and performance.

The article clearly and well-documented that the use of miniaturisation of the amplicon and metagenome library by the protocols used and well described therein reduces chemical and plastic costs, which is the primary focus of the reviewed article.

In addition to what has been described above, I agree with the decision to further optimise the PowerSoil Pro HT DNA Kit, as the data reported from extraction of samples using the above kit are very good, making the method a candidate for optimisation.

The miniaturised protocol was found to be cost effective and its optimisation is detailed in file S3.

The Materials and Methods section is well written. The protocols presented by the authors are well documented and confirmed. I have no corrections to make for this section, as the authors have documented it very well.

The discussion section can be implemented, below we list the points that I think need to be discussed further. All the data and results discussed in the article are deposited and accessible to the user, who can clearly understand the protocols used and the data generated.

The study presents the results of primary scientific research in a clear and meticulous manner, with results not reported in other previously published articles. Experiments, statics and analyses have been carried out to a high and meticulous technical standard, with the steps described and justified. All data and methods used are user-friendly.

The authors have presented an article that adheres to appropriate reporting guidelines and community standards for data availability. Applicable standards of experimental ethics and research integrity are met.

The manuscript is written in understandable standard English and I believe it can be accepted for publication after some minor possible additions or insights.

My additions and insights are listed below:

- Implementation within the abstract of the sequencing topic, as it is not sufficiently deepened.

- Deepening of the discussions, with primary focus on line 225-226, specifying the possible use of the protocols presented in a diagnostic context, given the average sequencing depth of 5 Gbp. On a practical level, in which contexts and which clinical/laboratory questions can I answer with this method, given an average sequencing depth of 5 Gbp?

- Implementation and deepening of the conclusions, specifying in concrete terms the possibility of using the protocols presented, given the large number of samples (2000), to achieve savings.

- Possible feasibility of the method in small laboratories, given the high cost of I.Dot instrumentation.

I would like to reiterate that my suggestions are only suggestions and should in no way affect the publication of the article as it is suitable for publication given the very high standards to which it has been written and documented.

I take this opportunity to congratulate the authors as the article is of a truly high scientific standard.

6. PLOS authors have the option to publish the peer review history of their article (what does this mean?). If published, this will include your full peer review and any attached files.

Reviewer #1: No

Reviewer #2: No

---

## [Editor Report · Acceptance letter]

26 Mar 2024

PONE-D-23-28692 

PLOS ONE

Dear Dr. Albertsen, 

I'm pleased to inform you that your manuscript has been deemed suitable for publication in PLOS ONE. Congratulations! Your manuscript is now being handed over to our production team.

Kind regards, 

on behalf of

Professor Vittorio Sambri 

Academic Editor

PLOS ONE